

# A methodological showcase: utilizing minimal clinical parameters for early-stage mortality risk assessment in COVID-19-positive patients

Jonathan K. Yan

Sage Hill School, Newport Coast, CA, USA

## ABSTRACT

The scarcity of data is likely to have a negative effect on machine learning (ML). Yet, in the health sciences, data is diverse and can be costly to acquire. Therefore, it is critical to develop methods that can reach similar accuracy with minimal clinical features. This study explores a methodology that aims to build a model using minimal clinical parameters to reach comparable performance to a model trained with a more extensive list of parameters. To develop this methodology, a dataset of over 1,000 COVID-19-positive patients was used. A machine learning model was built with over 90% accuracy when combining 24 clinical parameters using Random Forest (RF) and logistic regression. Furthermore, to obtain minimal clinical parameters to predict the mortality of COVID-19 patients, the features were weighted using both Shapley values and RF feature importance to get the most important factors. The six most highly weighted features that could produce the highest performance metrics were combined for the final model. The accuracy of the final model, which used a combination of six features, is 90% with the random forest classifier and 91% with the logistic regression model. This performance is close to that of a model using 24 combined features (92%), suggesting that highly weighted minimal clinical parameters can be used to reach similar performance. The six clinical parameters identified here are acute kidney injury, glucose level, age, troponin, oxygen level, and acute hepatic injury. Among those parameters, acute kidney injury was the highest-weighted feature. Together, a methodology was developed using significantly minimal clinical parameters to reach performance metrics similar to a model trained with a large dataset, highlighting a novel approach to address the problems of clinical data collection for machine learning.

# INTRODUCTION

Machine learning has been used extensively in health care, demonstrating a promising performance in predicting the outcome of diseases, including Alzheimer's disease (*Li et al., 2021*), cancer (*Tran et al., 2021*), diabetes (*Oikonomou & Khera, 2023*), and cardiovascular diseases (*Awan et al., 2018*). It is commonly believed that the accuracy of the ML relies on having access to extensive datasets for constructing models (*Cirillo & Valencia, 2019*; *Ching*

Corresponding author
Jonathan K. Yan,
maymayjon2@gmail.com

*et al., 2018*). However, data generated from patients is often costly and time-consuming. For example, in the emergency department, sparse features are common problems of electronic medical record data (*Wells et al., 2013*). Sparse features, characterized by zero values significantly larger than non-zero features, can significantly impact computing memory and diminish the model's ability to generalize. Particularly in small datasets, the substantial noise within sparse features can hinder model training, preventing convergence (*Heinze, 2006*; *Chen et al., 2023*). Recently, systematic approaches have been applied to build a prediction model to address this question, including a Random Forest (RF) for missing values, k-means for imbalanced data, and principal component analysis (PCA) for sparse features (*Chen et al., 2023*). Therefore, proving a methodology that similar effectiveness can be reached using minimal features compared to large datasets is essential to improve future applications to use ML in medicine.

Coronavirus disease 2019 (COVID-19) caused by the virus SARS-CoV-2 is a crucial disease that often affects the respiratory system (*Lu et al., 2020*). In January 2020, the global pandemic situation had been declared by the World Health Organization (WHO) (*Shi et al., 2020*). As of November 2023, COVID-19 had resulted in more than 6.9 million deaths worldwide, according to WHO. Despite the fact that vaccines for COVID-19 had been provided in the majority of the world and was effective for the prevention of COVID-19, delayed vaccination in underdeveloped countries or people who are against vaccination, and immune escape by variants will still result in COVID-19 infection (*Dunkle et al., 2022*; *Eyre et al., 2022*). The COVID-19 disease could progress in a variety of ways, from mild symptoms that go away on their own to serious and fatal side effects like acute respiratory distress syndrome (ARDS) and multi-organ failure (*Gautret et al., 2020*). It had been demonstrated that early therapies such antiviral drugs could stop clinical deterioration in COVID-19 patients (*Mathies et al., 2020*). Consequently, there is a significant need for early identification of potential severe outcomes in COVID-19 positive patients to enable prompt measures that can help reduce the risk of critical complications associated with the infection.

The pandemic has offered valuable datasets of clinical information linked to patient outcomes, providing a unique opportunity to develop a methodology using minimal clinical data for prediction. Recently, the potential of computational approaches proved to be significant in the diagnosis of COVID-19 at earlier stages. *Yao et al. (2020)* proposed a machine-learning-based model to detect the severity of COVID-19. The level of severity of COVID-19 was demonstrated by support vector machine (SVM) with 32 features using data from 137 COVID-19 patients. They further screened these 32 features for inter-feature redundancies. The final SVM model was trained using 28 features and achieved the overall accuracy 81% (*Yao et al., 2020*). In another study, a predictive model with top 20 features including baseline radiological, laboratory, and clinical data was reported. For mortality endpoint, the top model yielded an 80% accuracy using all features with balanced random forest (*Aljouie et al., 2021*). Moreover, chest CT pictures had been used for the training and prediction for COVID-19 outcome and become a valued feature of the assessment of patients (*Mei et al., 2020*). Therefore, although various studies have been performed using ML to predict the outcome for COVID-19 positive patients, very few studies have explored

the uses of the minimal clinical parameters at early stage of the infection to assess mortality risk.

In this study, a dataset encompassing over 1,000 patients was initially utilized for ML training purposes. To enable early assessment, only data from routine lab tests were included, excluding factors that signify patients are already in a severe condition. The minimal set of six clinical parameters was identified to assess COVID-19 outcomes at an early stage, demonstrating comparable accuracy to the use of 24 features.

## MATERIALS & METHODS

### Dataset

The dataset used in this study was from the Cancer Imaging Archive acquired at Stony Brook University from patients who tested positive for COVID-19 (*Saltz et al., 2021*). This collection includes associated clinical data for each patient. The clinical data consists of diagnoses, procedures, lab tests, COVID-19 specific data values (*e.g.*, intubation status, symptoms at admission) and a set of derived data elements, which were used in analyses of this data. The clinical data was stored as a set of csv files which comply with OMOP Common Data Model data elements.

### Data preprocessing

In the original dataset, each patient's features mainly consisted of true or false statements. All categorical features were binary encoded of features in order for the ML model to work properly. The dataset consists of many values that were labeled as ''NA''. These null values were removed from the dataset and the KNNImputer algorithm was used to fill in these missing values. The KNNImputer algorithm uses the k-nearest neighbors method to observe trends in the rest of the dataset and predict a value that best fits the missing value. The variables that did not require binary encoding were left in their raw integer state. However, the patients' ages were rounded to 38, 66, and 82 depending on which one they were closest to. Before preprocessing, this dataset originally contained 105 different columns of data and most of the features were removed as they contained over 30% null values so they would not be an accurate predictor. In this study, early detection is an essential part in helping to lower COVID-19 morality rates. However, in order to predict the outcome of a patient in the early stages of the disease, all the data that contained hospitalization had to be removed. As a result, the patient's ICU status, length of stay, and ventilation during hospitalization were dropped. In addition, the patients ID and their COVID-19 status were also dropped because all patients in the dataset were positive. Eventually, 30 different clinical features were used for this study.

### Machine learning algorithm selection

In the process of preparing the data for algorithm selection, patient results were isolated into a separate variable. Subsequently, the remaining 24 features underwent an 80% split into testing and training datasets. The objective was to identify the machine learning model that would yield the highest performance with the given data.

Various machine learning algorithms were then employed and assessed for their efficacy:

1. Support vector machine (SVM): SVM, a supervised machine learning algorithm, utilizes a hyperplane to delineate data with the maximum margin. It serves the dual purpose of classification and regression (*Shilton et al., 2005*).
2. K-nearest neighbors (KNN): KNN classification involves assigning unlabeled observations to the class of the most similar labeled examples, providing a proximity-based classification method (*Zhang, 2016*).
3. Random Forest (RF) Classifier: RF, an ensemble learning method, amalgamates multiple decision trees to make predictions. The training of models was executed using the training dataset, and accuracy was gauged by testing the trained model on the testing dataset (*Pedregosa et al., 2011*).
4. Logistic regression (LR) Model: LR, a regression analysis applicable to datasets with binary dependent variables, has found widespread use in epidemiology (*Bender, 2009*).

To enhance the performance of these machine learning algorithms, a grid search approach was implemented to fine-tune hyperparameters. These hyperparameters, additional parameters frequently adjusted by users, play a pivotal role in influencing prediction metrics (*Sah et al., 2022*).

This rigorous evaluation process aimed to determine the algorithm that exhibited the highest accuracy (ACC), ROC-AUC, and Matthew's correlation coefficient (MCC) scores, thus ensuring the selection of the most effective model.

1. Accuracy (ACC)

$$\text{Accuracy} = \frac{TP + TN}{TP + TN + FP + FN} \quad 0.0 <= \text{Accuracy} <= 1.0 \tag{1}$$

ROC-AUC

$$\text{ROC} - \text{AUC} = \int_0^1 TPR(FPR)dFPR \quad 0.0 <= \text{ROC} - \text{AUC} <= 1.0 \tag{2}$$

Matthew's Correlation Coefficient (MCC):

$$\text{MCC} = \frac{TP \times TN - FP \times FN}{\sqrt{(TP + FP)(TP + FN)(TN + FP)(TN + FN)}} \quad -1.0 <= MCC <= 1.0 \tag{3}$$

These metrics, derived from the confusion matrix variables (true positive, true negative, false positive, and false negative), provide a comprehensive evaluation of the model's performance.

## Feature importance evaluation

In the pursuit of determining the optimal number of features for maximal predictive performance, a comparative analysis was conducted between the random forest feature importance analysis algorithm and Shapley values obtained using XGBoost. The process involved multiple steps:

1. Random Forest feature importance: The weight of each feature was initially determined using the random forest feature importance analysis algorithm. This analysis assessed the relative importance of features based on their position within decision trees. Features at the top of the tree, with higher depth, were deemed more impactful in

determining final predictions. The expected fraction of sample contributions, combined with impurity reduction through splitting, generated a standardized measure of each feature's predictive capability (*Pedregosa et al., 2011*). The mean decrease in impurity (MDI), an aggregated metric, was utilized for subsequent feature selection (*Louppe, 2014*).

2. Shapley values using XGBoost: Shapley values, rooted in game theory and proposed by *Shapley (1953)*, were employed to assess the average marginal contribution of each feature to the results. XGBoost, an implementation of gradient boosted decision trees (*Parsa et al., 2020*), facilitated the computation of Shapley values.

3. Comparison and selection of features: The six features with the highest weights from the random forest feature importance analysis were juxtaposed with the six features possessing the highest Shapley values. Features exhibiting superior metrics were then used to construct a refined dataset, subsequently employed to train a new model.

This rigorous analysis, depicted in Fig. 1, aimed to identify a subset of features that, when employed in model training, would yield superior predictive performance.

## RESULTS

In this section, a comprehensive analysis of the results is presented. Leveraging 24 features initially, various machine learning algorithms were employed, with Logistic Regression and Random Forest emerging as the top performers.

Using the 24 features, it was shown that the Support Vector Machine (SVM) produced an accuracy of 0.8596, ROC-AUC score of 0.5, and an MCC score of 0. The K-Nearest Neighbors (KNN) classifier produced an accuracy of 0.8681, ROC-AUC score of 0.6190, and an MCC score of 0.3284. The Random Forest classifier produced an accuracy of 0.9106, ROC-AUC score of 0.7198, and an MCC score of 0.5744. The logistic regression classifier produced an accuracy of 0.9191, an ROC-AUC score of 0.7755, and an MCC score of 0.6322 (Fig. 2). The logistic regression and RF models obtained a better accuracy than the SVM and KNN classifiers after hyperparameter tuning. The higher accuracy of the RF classifier suggests that it was able to capture the underlying patterns and relationships within the dataset effectively. This is likely due to the ensemble nature of RF, which combines multiple decision trees to make more accurate predictions. The higher performance metrics of the logistic regression classifier suggests that it was able to capture the underlying patterns and relationships within the dataset effectively.

Furthermore, to identify minimal clinical parameters that could be associated with the mortality of COVID-19, the random forest feature importance analysis algorithm was employed. This approach allowed for the determination of the relative importance of each feature in observing outcomes (Fig. 3A). Among the 24 features, the six most important features were acute kidney injury status, serum glucose level, age, high troponin level (above 0.01), acute hepatic injury, and blood oxygen level measured by pulse oximeter (under 90). These features have an importance of 0.15, 0.13, 0.07, 0.06, 0.04, and 0.04, respectively. Using XGBoost, the Shapley values of all features were determined (Fig. 3B). The six features with the highest Shapley values were acute kidney injury, age, serum glucose

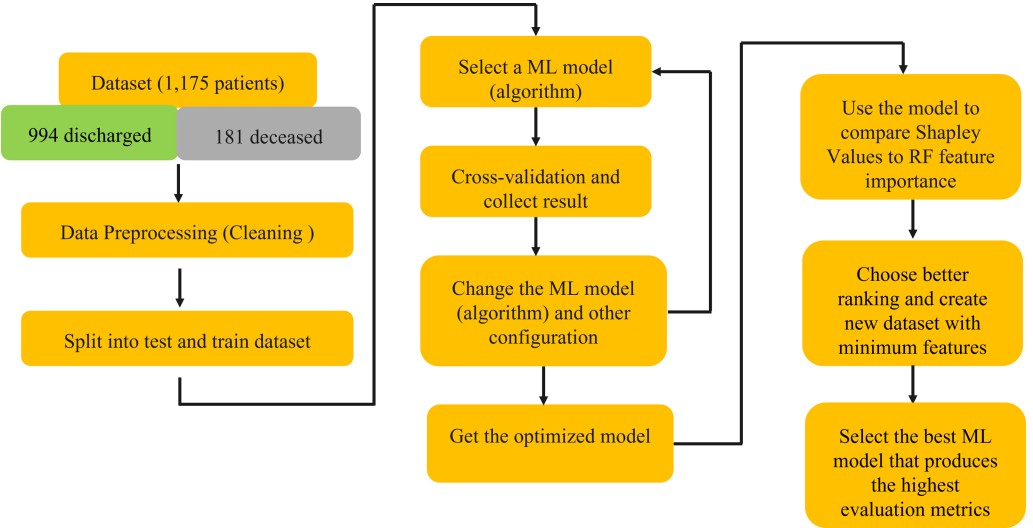

**Figure 1** **Schematic explanation of the methods using in this study.**

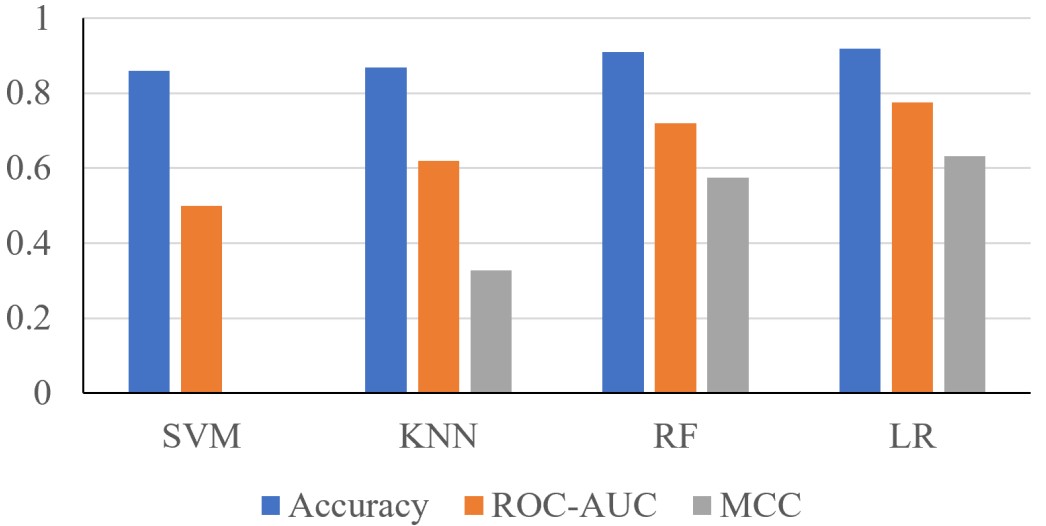

■ Accuracy  ■ ROC-AUC  ■ MCC

**Figure 2** **Performance metrics of four prediction algorithms trained using all features.** The *x*-axis contains the four ML algorithms. The *y*-axis contains the values of these metrics.

level, pulse oximeter (under 90), aspartate (over 40), and MAP (above 90). The Shapley values for these features were 1.5, 1.1, 0.67, 0.62, 0.52, and 0.5. Next, Shapley values of each feature were compared with their random forest feature importance. To determine which list of features to train the final model with, logistic regression models were trained with the top six features from the Shapley values and the Random Forest feature importance (Fig. 4A). The model that was trained with the Shapley value features produced an accuracy of 0.9064, a ROC-AUC score of 0.7808, and an MCC score of 0.5931. The model trained with the RF features produced an accuracy of 0.9149, a ROC-AUC score of 0.7857, and an

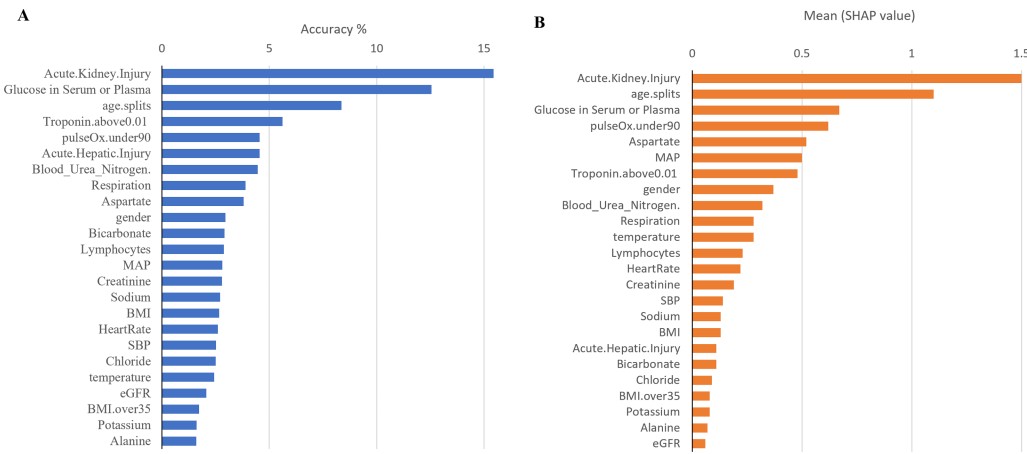

**Figure 3** **Comparison between Shapley values and RF feature importance.** (A) The predicted weight of each feature using the random forest feature importance algorithm. (B) The predicted Shapley value of each feature using XGBoost.

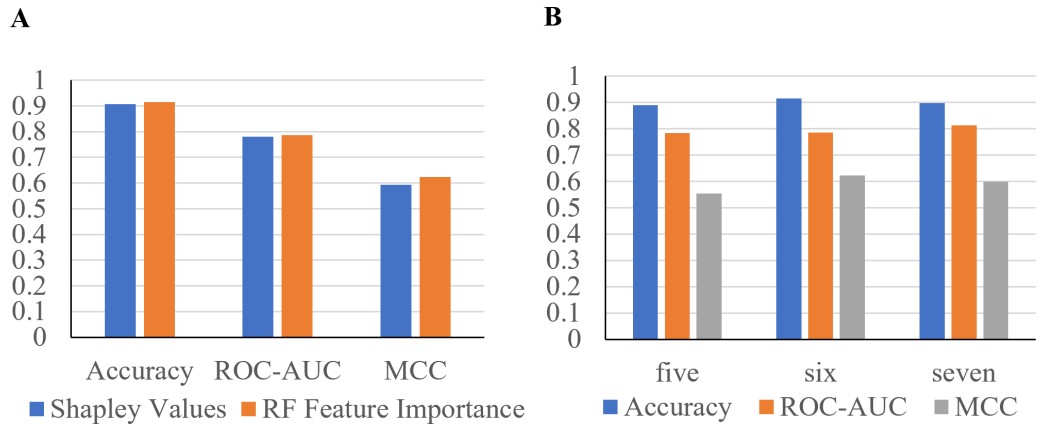

**Figure 4** **Performance metrics of logistic regression models trained with the six most important features.** (A) The features were determined using Shapley values and random forest feature importance analysis. (B) Performance metrics of logistic regression models trained with the five and seven most important features. Features were determined with the RF feature importance analysis algorithm.

MCC score of 0.6225. As a result, the features that were determined using the RF feature importance analysis algorithm were better suited for this study.

To confirm that six features can produce the highest performance metrics, two Logistic Regression models were trained with five or seven features (Fig. 4B). The model trained with the top five most important features produced an accuracy of 0.8894, a ROC-AUC score of 0.7835, and a MCC score of 0.5534. The model trained with seven features produced an accuracy of 0.8979, a ROC-AUC score of 0.8138, and a MCC score of 0.5987. The model trained with six features produced the highest performance metrics, proving that it is the best option.

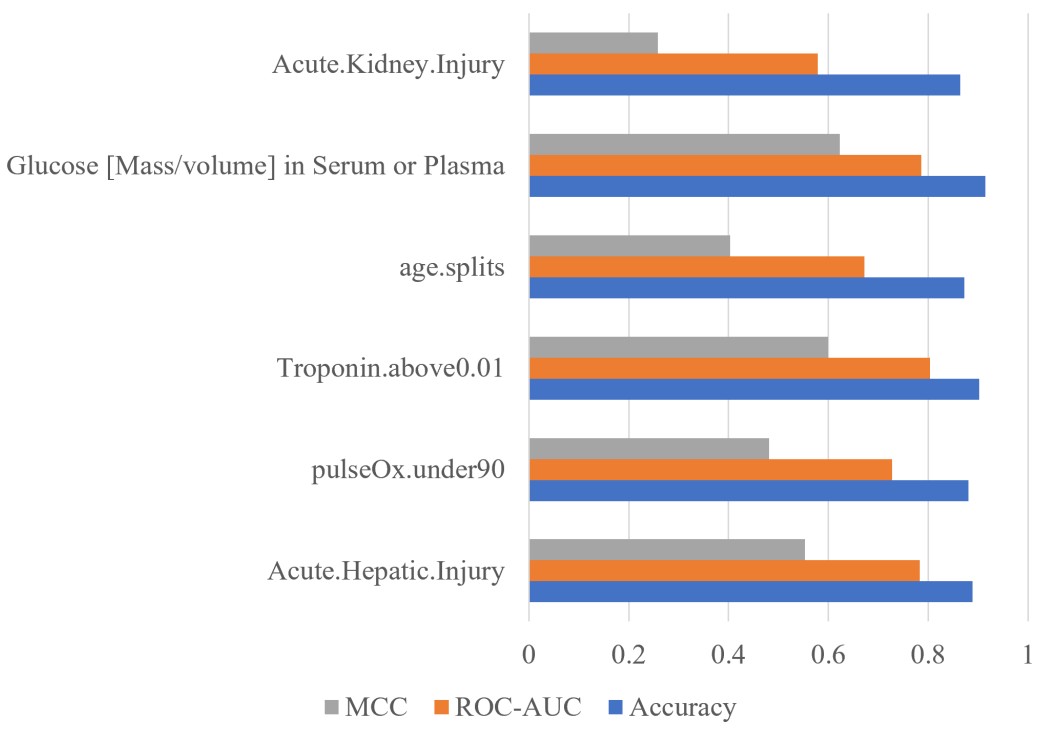

**Figure 5** **Acute kidney injury is the most important feature determined by using ablation experiment.** Performance metrics of logistic regression models trained during an ablation experiment executed with the six most important features determined using the RF feature importance algorithm. The *y*-axis contains the name of the feature that was removed. The *x*-axis contains the values of the performance metrics.

To confirm the importance of each feature, an ablation experiment was conducted. Using the logistic regression ML algorithm, the performance metrics were gathered for models trained with the six most important features determined with the RF feature importance analysis algorithm (Fig. 5). For each of six tests, one of the six features was removed. When acute hepatic injury was removed the accuracy was 0.8894, the ROC-AUC score was 0.7835, and the MCC score was 0.5534. Without saturation of oxygen in the blood (under 90) the model produced an accuracy of 0.8809, a ROC-AUC score of 0.7279, and a MCC score of 0.4814. When troponin above 0.01was removed, the accuracy became 0.9021, the ROC-AUC score became 0.8036, and the MCC score became 0.5998. When age was removed, the accuracy was 0.8723, the ROC-AUC score became 0.6722, and the MCC score became 0.4027. When serum glucose level was removed, the accuracy became 0.9149, the ROC-AUC score became 0.7857, and the MCC score became 0.6225. When the acute kidney injury was removed, the model produced an accuracy of 0.8638, a ROC-AUC score of 0.5785, and a MCC score of 0.2583. Without acute kidney injury, the model's performance decreased the most, meaning that acute kidney injury is the most important feature.

Finally, to confirm logistic regression is the best model to be used on the six most important features, each model was reassessed with the smaller set of data (Fig. 6A). Using six features, the SVM classifier produced an accuracy of 0.8596, ROC-AUC score of

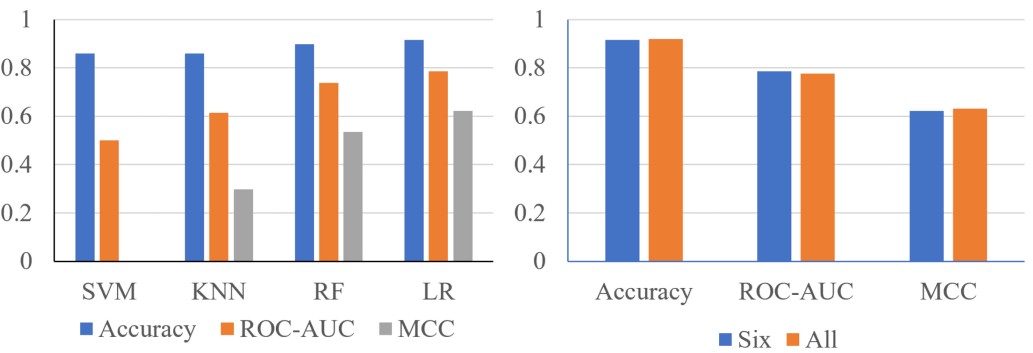

**Figure 6** **The metrics produced are similar, using only six features compared to 24 features.** (A) Performance metrics of four prediction algorithms trained using the highest weighted six features determined with the random forest feature importance algorithm. (B) Performance metrics for logistic regression models trained with the most important six features and all 24 features.

0.5000, and an MCC score of 0.0000. The KNN classifier produced an accuracy of 0.8596, ROC-AUC score of 0.6141, and an MCC score of 0.2981. The random forest classifier produced an accuracy of 0.8979, a ROC-AUC score of 0.7378, and a MCC score of 0.5359. The logistic regression model produced a surprising accuracy of 0.9149, a ROC-AUC score of 0.7857, and a MCC score of 0.6225. The logistic regression model was able to produce the best performance metrics confirming that it is the best ML algorithm to use. Therefore, the performance metrics produced are similar using only six features compared to 24 features (Fig. 6B).

## DISCUSSION

This study addresses the challenges posed by limited data availability in machine learning applications within clinical settings, particularly emphasizing the need to develop methodologies that can offer valuable insights into patient outcomes based on a restricted set of clinical parameters. The article takes advantage of the publicly available dataset of the COVID-19 patients and aims to observe potential mortality indicators in the early stages of COVID-19 infection. The study uses features that are obtained through routine lab test results while excluding features that indicate the patient is already in a severe condition. Employing a logistic regression model, the research achieved an accuracy of 0.9191, a ROC-AUC score of 0.7755, and a MCC score of 0.6322 by using 24 clinical parameters. The investigation found that the six most highly weighted clinical parameters were acute kidney injury, glucose level, age, troponin, oxygen level, and acute hepatic injury. The final logistic regression model, trained with these six features, produced an accuracy of 0.9149, a ROC-AUC score of 0.7857, and an MCC score of 0.6225. The study outlines the feasibility of utilizing minimal clinical parameters for COVID-19 outcome estimation. It highlights the potential for creating ML models in future using a handful of key features, showcasing their potential in yielding precise predictions in studies concerning human health.

The study contextualizes its findings within the broader landscape of machine learning applications in healthcare, particularly in predicting COVID-19-related mortality. For

example, one study used 15 parameters including gender, age, blood urea nitrogen (BUN), creatinine, international normalized ratio (INR), albumin, mean corpuscular volume (MCV), white blood cell count, segmented neutrophil count, lymphocyte count, red cell distribution width (RDW), and mean cell hemoglobin (MCH) along with a history of neurological, cardiovascular, and respiratory disorders to predict patient outcomes with a sensitivity of 70% and a specificity of 75% (*Jamshidi et al., 2021*). Despite the limited dataset, the study demonstrates that even a minimal set of clinical parameters can yield insights into mortality risks associated with COVID-19, offering a pragmatic approach for early-stage risk assessment.

Understanding the multifaceted effects of the COVID-19 pandemic on mortality across different population levels presents a significant challenge for public health research. Two recent publications demonstrated the direct COVID-19 contribution to excess mortality exceeds 100% in the oldest age groups (*Lee et al., 2023*; *Riou et al., 2023*). The age groups affected most by excess mortality were those over 70 years of age. Based on these studies, patients aged over 74 have been selected and performance metrics on a logistic regression model trained with all 24 features of these patients have been gathered. The model achieved an accuracy of 0.8113, an ROC-AUC score of 0.7680, and a MCC score of 0.5553. A logistic regression model trained with the top six features of these patients produced an accuracy of 0.7736, an ROC-AUC score of 0.7092, and a MCC score of 0.4538. These two models still produced similar performance metrics, meaning that using minimal clinical parameters, one can still observe mortalities directly related to COVID-19. The comparative performance of these models suggests that a select few clinical parameters can indeed provide valuable insights into mortality risk, highlighting a potential area for further research. This approach underscores the importance of identifying key indicators within limited data sets, thus contributing to a better understanding of COVID-19's impact while acknowledging the limitations inherent in observational studies and the need for careful interpretation of these correlations.

Using the RF feature importance analysis algorithm, Shapley value analysis, and ablation experiment, this study has identified a strong association between acute kidney injury and increased mortality among COVID-19 patients. This aligns with previous findings indicating that acute kidney injury significantly elevates the risk of mortality in COVID-19 patients, with a risk ratio (RR) of 4.6 for mortality compared to those patients with COVID-19 but without acute kidney injury (*Fu et al., 2020*). Pathological studies using both live kidney biopsies and autopsy samples demonstrate that acute tubular injury is the most commonly encountered findings in COVID-19 patients (*Sharma et al., 2021*). It has been well acknowledged that COVID-19 patients with high blood glucose level will be more likely to transform into severe and fatal cases (*Cai et al., 2020*; *Coppelli et al., 2020*; *Sardu et al., 2020*). Surprisingly, the ablation experiment revealed that the exclusion of the glucose level feature did not significantly affect the model's outcome, suggesting a nuanced relationship between glucose levels and COVID-19 mortality that needs further investigation. This research underscores the correlation between acute kidney injury and higher mortality rates in COVID-19 patients, emphasizing the critical need for monitoring

of this condition in affected individuals. Further investigations need to be performed to validate whether these parameters can be used as definitive predictors of mortality.

This study signifies a crucial advancement in the development of methodologies aimed at utilizing a minimal dataset to identify and leverage the most significant features for analysis. This approach has demonstrated the potential to achieve accuracies comparable with those derived from more extensive datasets. Future efforts should focus on the thorough validation of this approach across a variety of patient demographics to ascertain its robustness and generalizability for predictive analytics in healthcare settings.

## CONCLUSIONS

In conclusion, this research has successfully leveraged various machine learning models to develop a novel methodology involving the use of minimal clinical parameters to generate a model with similar performance to a model trained with all features. This study also showed that parameters such as acute kidney injury and glucose level are highly correlated with high COVID-19 mortalities. Notably, the model relies on a mere six clinical parameters, achieving an accuracy exceeding 90%. This groundbreaking achievement holds significant potential by allowing for clinical data gathering to be more cost efficient and effective.

## ACKNOWLEDGEMENTS

The author would like to thank Dr. Sun Yu at Cal Poly Pomona for their insightful suggestions.

### Funding
The authors received no funding for this work.

### Competing Interests
The author declares that they have no competing interests.

### Author Contributions
- Jonathan K. Yan conceived and designed the experiments, performed the experiments, analyzed the data, performed the computation work, prepared figures and/or tables, authored or reviewed drafts of the article, and approved the final draft.

### Data Availability
The COVID-19-NY-SBU dataset is available at: https://wiki.cancerimagingarchive.net/pages/viewpage.action?pageId=89096912.

The reproducible code is available at GitHub and figshare:

– https://github.com/Maymayjon/COVID19PatientMortalityPredictor.

– Yan, Jonathan (2024). COVID19PatientMortalityPredictor-1.zip. figshare. Journal contribution. https://doi.org/10.6084/m9.figshare.25557645.v1.

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
