# Peer review of "A methodological showcase: utilizing minimal clinical parameters for early-stage mortality risk assessment in COVID-19-positive patients"

_PeerJ Computer Science, doi:10.7717/peerj-cs.2017_

## Round 0.1 · original submission · Major Revisions

The reviewers have substantial concerns about this manuscript. The authors should provide point-to-point responses to address all the concerns and provide a revised manuscript with the revised parts being marked in different color.

**Language Note:** PeerJ staff have identified that the English language needs to be improved. When you prepare your next revision, please either (i) have a colleague who is proficient in English and familiar with the subject matter review your manuscript, or (ii) contact a professional editing service to review your manuscript. PeerJ can provide language editing services - you can contact us at copyediting@peerj.com for pricing (be sure to provide your manuscript number and title). – PeerJ Staff

Reviewer 1 ·

Basic reporting

This article discusses the issue of data scarcity in machine learning especially for the health science. The author showcased a methodology to build up a model to predict the outcome of COVID-19-positive patients with minimal clinical parameters at an early stage. And discussed the key features of clinical parameters. The overall manuscript was insufficient persuasiveness and insufficient details.

In section Introduction, paragraph 2 and 3, the past tense should be used when introduce published work. In paragraph 3, the author should create a new paragraph to describe their study and model design concept. Besides, the author should cite more work of sparse data modeling.

The four Figures were bit blurry, please replace them with a clearer picture. Figure 2-4 should be merged into one picture. The font size should be adjust to similar size with the main text. The figure should give some description of the results.

Experimental design

Ablation experiments should be taken to study the importance of the different features. Pathological inferences should be provided to analysis in the results and discussion section.

The full name of variable abbreviation (TP, TN, FP, etc. ) should be given in the equation for accuracy calculation, and similarly for other abbreviations. Higher order and deeper level indicators should be provided to give more detailed analysis.

Validity of the findings

In Section Discussion, the first paragraph was duplicated with section Results. Line spacing of the first paragraph was not correct. The format throughout the article need to be carefully revised.

Reviewer 2 ·

Basic reporting

The paper titled "A methodological showcase: Utilizing minimal clinical parameters to predict mortality of COVID-19 positive patients at early-stage" discussed the development of a model to predict the outcome of COVID-19-positive patients using minimal clinical parameters. The author highlights the challenge of limited data in machine learning and the need to develop methods that can achieve accurate predictions with minimal data. The study focuses on predicting the mortality of COVID-19-positive patients at an early stage using only routine lab test results, excluding factors that indicate severe situations. A neural network model was built with over 90% accuracy when combining 24 clinical parameters. The study identifies six key clinical parameters, including acute kidney injury, glucose level, age, troponin, oxygen level, and acute hepatic injury, which have the most weight for prediction accuracy. The final model using these six features achieved an accuracy of 88% using a random forest classifier and 91% using a logistic regression model. The study concludes that minimal clinical parameters can be used for COVID-19 outcome prediction, highlighting the importance of developing models with few key features for accurate predictions in human health-related studies.

I have a few concerns that would like the authors to address:
1.Please provide MCC as metric to evaluate the performance of your classification task.
2.I recommend to perform the Shapley Value by using XGBoost approach, and compare the importance against RF model with more explainable information.

Experimental design

As above

Validity of the findings

As above

Additional comments

As above

Reviewer 3 ·

Basic reporting

The author clearly stated the importance of data scarcity issues in health sciences and the necessity of utilizing the so-called minimal parameters to reach on-par model performance. The author tried to tackle the problem of COVID-19 mortality prediction using the minimal parameters approach and stated the experiments in a relatively clear English. However, there are several major issues worth extensive attentions (failed to meet basic report guidelines):
1). In the abstract, the author claimed that “A neural network model was built with over 90% accuracy when combining 24 clinical parameters.” (Line 23-24). However, no details about neural-network based model given or mentioned in the later main text, only SVM/RF/KNN/simple logistic regression were later involved. This is a major conflict.
2). The author should avoid using first-person narrative (“I”) in the methods and results sections in a research paper: Line 97, Line 142 and Line 154.
3). Further references required at the following positions:
a. Line 54, “Therefore, there are still over 1million confirmed cases…”
b. Line 165, “Although AI based predictions of COVID-19…”
c. Line 122-128, the determination of RF feature importance should cite corresponding refs.
4). The author should stick to professional English more instead of daily oral language. For example,
a. Line 96, “I changed all of them to 0’s and 1’s” could be rephrased as "all categorical variables/features were binary encoded/labeled of features" (and again as mentioned above, first person narrative using "I" should be avoided here in the methods section)
b. Line 76 and Line 165, the author is recommended to refrain from using the word AI (which is more suited for generative ML works) for simple binary classification ML models.
c. Line 178, “this study” is confusing immediately after a cited research, the author should rephrase as “In our/my studies…”
d. Line 187, the author mentioned the application (APP) without giving details on how such APP worked (is it just a jupyter notebook as indicated in the original code repo on the GitHub?).

Experimental design

The author raised an interesting ML prediction topic about using the minimal parameters approach for COVID-19 mortality prediction (or early risk prediction). However, given the data generation process is purely out of author’s control (open access data) and data size is very limited (~1385 total data rows), the study is more suited to be framed under the observational studies (causal inference) rather than a pure predictive study. In addition to this main directional mismatch, there are other major issues worth further investigation:
1). The definition of mortality: the author take last.status as the final target feature for the whole modeling, however, the author failed to investigate if such last.status is the direct outcome of COVID-19 or from other hidden status, such as inherent fundamental diseases if the whole purpose of the article is to investigate early risk call-out for COVID-19.
2). The data preprocessing:
a). the author first mentioned all null values were removed (Line 97-98) then later also mentioned KNNImputer to fit the missing values without mentioning really using it or not in the research. This is conflicting with each other.
b).Also, the author failed to provide details on categorical variables translation, only mentioning 0-1 binary encoding for binary value variables, what about other categorical variables with value options larger than 2?
c).More importantly, the author mentioned the original dataset contains 30 different features of COVID-19 patients (Line 100), however, upon checking original data, it actually contains more than 100 features of patients, the author failed to address the such huge mismatch.
3). Metric definition: the author only used accuracy as the sole metric for tracking during the whole research but failed to justify its sole usage. Why F1-score, AUCROC, which are more comprehensive metrics in the binary classification problems, not applied here in the research? The author needs to provide more details on that. Additionally, for Line 119, Accuracy (ACC) is 0.0 <= Accuracy <= 1.0, not just < .
4). The author mentioned using SVM/RF/KNN/simple logistic regression models, but without mentioning any hyper-parameters tuning, particularly for the RF model. This is not recommended for experiments replication.
5). The author failed to demonstrate why in the end only 6 features were chosen for the final model. Need further supportive evidence to justify the final 6-feature based conclusion.

Validity of the findings

The author concluded that with the minimal parameters approach introduced in the article, only 6 features are required to reach >90% accuracy for a COVID-19 mortality prediction model. However, such conclusion is very hasty without rigorous investigation:
1). The author claimed that “This study demonstrated that acute kidney injury is the most important clinical parameter that is highly associated with patient mortality…”, without any confounding effects control and regulation, with no clear definition of mortality: because of COVID-19 directly or indirectly? The author failed to demonstrate any controlling or regulation mechanisms for the underlying data to reach the conclusions provided in the article.
2). The author claimed that “This study represents a novel approach in the early prediction of COVID-19 outcomes…”. However, given the limited data size and no data generation process in control, this research is not suited for a prediction task but rather an observational study to uncover the potential correlations between acute kidney injury/blood gluocose level and COVID-19 mortality/final outcomes (if any).
3). Given internal contextual conflicts, such as mentioning neural networks but without really using it, the feature counts mismatch (claiming 30 features but in original data more than 100 features exist), no details provided on data preprocessing or hyperparameters of ML models, the conclusions generated in the end is not robust or sound.
In the end, It is worth mentioning that author’s efforts to include all original data and original code repo on the GitHub are greatly appreciated.

---

## Round 0.2 · Major Revisions

While the current revised manuscript addressed some reviewers' concerns, one reviewer is still questioning the conclusions of the manuscript. Authors should seriously address the remaining concerns and provide point-to-point responses to them.

Reviewer 1 ·

Basic reporting

The issues proposed in the last manuscript have been revised, and some details have also been addressed. This version facilitates readers to better convey the author's viewpoint.

Experimental design

no comment

Validity of the findings

no comment

Reviewer 2 ·

Basic reporting

I have no further comments on this research paper.

Experimental design

I have no further comments on this research paper.

Validity of the findings

I have no further comments on this research paper.

Additional comments

I have no further comments on this research paper.

Reviewer 3 ·

Basic reporting

Pass. The author clearly stated the importance of data scarcity issues in health sciences and the necessity of utilizing the so-called minimal parameters to reach on-par model performance. The author cleared the major concerns proposed in the previous review.

Experimental design

The author cleared minor concerns, such as more detailed data preprocessing, RF based features selections etc. However, the major concern that "given the data generation process is purely out of author’s control (open access data) and data size is very limited (~1385 total data rows), the study is more suited to be framed under the observational studies (causal inference) rather than a pure predictive study. " is not addressed and the whole structure of manuscripts is largely unchanged. Based on the limited sample size and without data generation process insight, one can hardly draw solid conclusion that using acute kidney injury, glucose level can "predict" each covid-19 patient's mortality. One can only conclude that those parameters such acute kidney injury, glucose level are highly correlated with high covid-19 mortalities and worth further investigation.

Validity of the findings

the author has demonstrated his efforts to resolve some of the contextual conflicts (such as removing neural networks wording, giving more details on the feature selection etc.). However, the major concern is still, that the generation process is outside author's control and there is no clear variable control or confounding variables control (citing other paper's results cannot form the conclusion in the manuscript directly) mechanism mentioned in the revised manuscript, the conclusion drawn from the manuscript is still largely flawed and can only show the correlation relationship, far from "predicting" covid-19 mortality.

---

## Round 0.3 · accepted · Accept

The author has addressed all concerns raised by reviewers.